# DSPO: Stable and Efficient Policy Optimization for Agentic Search and Reasoning

## Abstract

Enhancing LLMs with the ability to actively search external knowledge is crucial for complex and real-world tasks. Current approaches either rely on prompting to elicit the model's innate agent capabilities, or suffer from performance ceilings and collapse when applying RL to complex interactive tasks, leaving their true agentic potential untapped. To address this, we introduce **D**ynamic-filter **S**equence-level **P**olicy **O**ptimization (DSPO), an improved RL algorithm designed for robust agent training through sequence-level optimization and dynamic sample filtering. We train our model purely through RL to interleave multi-turn search and reasoning, obviating the need for supervised demonstration data. Across multiple QA benchmarks, our DSPO-trained 7B model improves over a comparable previous work by **34.1%**, and even outperforms the 14B model from previous work in complex multihop QA such as HotpotQA by nearly **9% relative**, maintaining exceptional training stability.

## 1 Introduction

Large Language Models (LLMs) (Brown et al., 2020; Touvron et al., 2023; Zhao et al., 2023) have demonstrated exceptional performance across a spectrum of specialized tasks, including math (Shao et al., 2024; Trinh et al., 2024; Yu et al., 2025), coding (Zheng et al., 2023; Yang et al., 2024), and creative writing (Chakrabarty et al., 2024; Marco et al., 2024). However, a fundamental limitation persists: their knowledge is inherently static, confined to the data on which they were trained. To overcome this knowledge cutoff, a dominant approach is to equip LLMs with search capabilities, transforming them into agents that can actively query external knowledge sources (Jin et al., 2025b). This ability is a prime example of tool-calling (Schick et al., 2023), where the model learns to interact with an external search tool to solve problems it cannot answer alone. Mastering this skill requires learning a complex, multi-step policy, framing the task as a sequential decision-making problem ideal for Reinforcement Learning (RL).

Unlike Supervised Fine-Tuning (SFT), which relies on costly static demonstrations and fails to teach exploration, RL provides a framework for LLMs to learn effective policies through trial-and-error (Chu et al., 2025). Consequently, value-free methods like Group Relative Policy Optimization (GRPO) (Shao et al., 2024) have become a dominant paradigm, prized for their simplicity and reduced memory overhead. However, despite its success in more constrained tasks, applying GRPO to the open-ended domain of interactive search reveals critical instabilities (Jin et al., 2025b; Yu et al., 2025; Cui et al., 2025; Liu et al., 2025). This fragility stems from two fundamental flaws. First, as identified by Zheng et al. (2025), GRPO's token-level objective is ill-posed when paired with a sequence-level reward, creating high-variance gradients that destabilize training. Second, the sparse rewards inherent to search tasks often yield sample groups with homogeneous outcomes (e.g., all successes or all failures), causing the advantage signal to collapse and providing no learning signal, which severely hinders sample efficiency (Yu et al., 2025; Liu et al., 2025).

To address these core challenges of instability and inefficient learning, we introduce **D**ynamic-filter **S**equence-level **P**olicy **O**ptimization (DSPO). Our algorithm synthesizes and refines key principles from recent policy optimization research. DSPO adopts the sequence-level optimization from GSPO (Zheng et al., 2025) to match the unit of optimization with the unit of reward. This aligns the optimization objective with the reward signal, fundamentally stabilizing the learning process for long-horizon reasoning tasks. Furthermore, DSPO incorporates a dynamic outcome-based filtering

Figure 1: An overview of the DSPO training loop. For a given query, the policy model generates a group of $G$ trajectories by interacting with the search environment. Each trajectory is assigned a sparse terminal reward. The **dynamic filter** discards groups with homogeneous outcomes and keep sampling until a batch is filled, ensuring that every training batch provides a effective advantage signal. Advantages are computed and used to update the policy model via sequence-level objective.

mechanism inspired by DAPO (Yu et al., 2025). This component actively constructs training batches from rollout groups containing both successful and unsuccessful outcomes for each prompt. It guarantees the advantage signal $\hat{A}_i$ to be effective and stable. By integrating these two components into a single, coherent framework, DSPO provides a stable and high-performance algorithm designed for complex, multi-turn search and reasoning tasks. Our model achieves a **34.1% relative improvement** over a leading 7B baseline (Jin et al., 2025b) and even surpasses its 14B counterpart (Jin et al., 2025a) on complex multi-hop benchmarks like HotpotQA, outperforming it by **nearly 9% relative** (0.613 vs. 0.563).

In summary, our main contributions are as follows:

- We propose DSPO, an improved RL algorithm that overcomes the core instability and sample-inefficiency issues in training agentic search models. It achieves this by unifying two key principles into a single cohesive framework: **sequence-level optimization** for robust policy updates and **dynamic outcome-based filtering** for a dense and effective learning signal.

- We demonstrate DSPO's substantial performance gains through rigorous benchmarking. Our 7B model achieves a **34.1% relative improvement** over a comparable 7B baseline and, more strikingly, outperforms its 14B counterpart on complex multi-hop QA, achieving a nearly **9% relative gain** on HotpotQA (0.613 vs. 0.563).

- We provide extensive empirical evidence for DSPO's superior training stability, showing it enables a stable learning trajectory. Crucially, the results are achieved using only a basic BM25 retriever, isolating the performance gains to the robustness of our algorithm.

## 2 RELATED WORK

### 2.1 RL FOR LLMS

The landscape of RL for LLMs has evolved rapidly, moving from foundational Reinforcement Learning from Human Feedback (RLHF) methods that use PPO and explicit reward models (Ouyang et al., 2022; Christiano et al., 2017; Schulman et al., 2017) to simpler, direct-optimization frameworks like DPO (Rafailov et al., 2023). A key shift towards value-free optimization is marked by Group Relative Policy Optimization (GRPO) (Shao et al., 2024), which simplifies training by deriving a reward signal from group statistics. However, GRPO's token-level objective is known to cause training instability (Liu et al., 2025; Cui et al., 2025), prompting several targeted improvements. GSPO addresses this by shifting to a sequence-level objective to match the unit of reward (Zheng et al., 2025), while DAPO tackles inefficient learning from sparse rewards with a dynamic outcome-based sampling mechanism (Yu et al., 2025). In a similar vein, GMPO stabilizes the token-level objective using a geometric-mean aggregation to reduce sensitivity to outliers (Zhao et al., 2025).

Despite these advances, we observed these algorithms still face challenges like training collapse or performance bottlenecks in our experiments. Building upon the aforementioned research, we propose our improved algorithm, synthesizing the principles of sequence-level optimization and dynamic filtering and filling into a unified algorithm to overcome the unique challenges of training autonomous search agents.

## 2.2 LLMs with Agentic Retrieval

To mitigate the static knowledge limitations of LLMs, RAG integrates external retrievers to dynamically incorporate evolving information (Lewis et al., 2020; Gao et al., 2023). Classic RAG frameworks employ dense retrievers to fetch relevant documents, which are then concatenated into the LLM's input for generation (Karpukhin et al., 2020). However, these approaches often rely on fixed pipelines, limiting autonomy in complex, multi-turn scenarios. Recently, research has evolved toward agentic paradigms, where LLMs act as autonomous agents capable of planning, searching, and reasoning iteratively. Frameworks like ReAct synergize reasoning and acting, enabling LLMs to interact with tools for tasks such as web navigation (Yao et al., 2023), while multi-agent systems, including AutoGen, facilitate collaborative workflows (Wu et al., 2024). Recent innovations emphasize agentic RAG and RL integration, where agents enhance retrieval through decision-making. Wu et al. (2025) introduce Agentic Reasoning, a framework integrating external tools for streamlined LLM reasoning. Some RL-integrated approaches (Jin et al., 2025b; Chen et al., 2025; Song et al., 2025) train LLMs to interleave reasoning and search using purely RL. The end-to-end paradigm internalizes agent capabilities and can avoid the engineering overhead of multi-agent frameworks. However, these methods still grapple with the training instability and performance limitation to the open-ended search domain. Our work directly confronts these bottlenecks. DSPO provides a robust and efficient training framework that ensures stable policy optimization, enabling LLMs to learn effective multi-turn search strategies.

## 3 Methodology

In this section, we first formulate the task of agentic search as a RL problem and review prior policy optimization algorithms, highlighting their limitations in this context. We then introduce our proposed algorithm, **D**ynamic-filter **S**equence-level **P**olicy **O**ptimization (DSPO), detailing its core components for training stability and training efficiency. Finally, we present the integrated training algorithm.

### 3.1 Preliminaries

**Policy Gradient Methods for LLMs.** Training LLMs via RL often employs policy gradient methods like PPO (Schulman et al., 2017), a popular algorithm for LLM alignment. It optimizes a policy $\pi_\theta$ by maximizing a clipped surrogate objective function using samples from an old policy $\pi_{\theta_{\text{old}}}$. The objective, averaged over tokens, is given by:

$$J_{\text{PPO}}(\theta) = \mathbb{E}_{x \sim \mathcal{D}, y \sim \pi_{\theta_{\text{old}}}(\cdot|x)} \left[ \frac{1}{|y|} \sum_{t=1}^{|y|} \min\left( r_t(\theta) \hat{A}_t, \text{clip}\left( r_t(\theta), 1 - \epsilon, 1 + \epsilon \right) \hat{A}_t \right) \right], \quad (1)$$

where $r_t(\theta) = \frac{\pi_\theta(y_t|x,y_{<t})}{\pi_{\theta_{\text{old}}}(y_t|x,y_{<t})}$ is the token-level importance ratio. However, PPO relies on a separately trained value model to estimate token-level advantages $\hat{A}_t$ via Generalized Advantage Estimation (GAE) (Schulman et al., 2015), introducing significant memory overhead and can be a source of instability.

To address this, GRPO (Shao et al., 2024) was proposed. GRPO eliminates the need for a value model by sampling a group of $G$ responses $\{y_i\}_{i=1}^G$ for a given prompt $x$. It then calculates the advantage of each response by normalizing its reward against the group's statistics. Like PPO, it optimizes the objective at the token level:

$$J_{\text{GRPO}}(\theta) = \mathbb{E}_{x \sim \mathcal{D}, \{y_i\}_{i=1}^G \sim \pi_{\theta_{\text{old}}}(\cdot|x)}$$

$$\left[ \frac{1}{G} \sum_{i=1}^G \frac{1}{|y_i|} \sum_{t=1}^{|y_i|} \min \left( r_{i,t}(\theta)\hat{A}_i, \text{clip}(r_{i,t}(\theta), 1 - \epsilon, 1 + \epsilon)\hat{A}_i \right) - \beta D_{KL}(\pi_\theta || \pi_{\text{ref}}) \right], \tag{2}$$

where $r_{i,t}(\theta) = \frac{\pi_\theta(y_{i,t}|x,y_{i,<t})}{\pi_{\theta_{\text{old}}}(y_{i,t}|x,y_{i,<t})}$, and the advantage for every token $y_{i,t}$ in a response $y_i$ is set to the same sequence-level value:

$$\hat{A}_{i,t} = \hat{A}_i = \frac{R_i - \text{mean}(R)}{\text{std}(R)}, \tag{3}$$

Crucially, all tokens within a given response $y_i$ share the same advantage $\hat{A}_i$, which is derived from the sequence-level reward.

**Agentic Search as a Markov Decision Process.** We model the iterative process of agentic search and reasoning as a sequential decision-making problem, formalized as a discrete-time, finite-horizon Markov Decision Process (MDP).

- **State ($s_t$):** At turn $t$, the state $s_t$ encodes the entire interaction history: the initial question $q$, all previously generated thoughts and search queries, and the retrieved evidence returned by the environment. Because search results must be interpreted and integrated into subsequent decisions, the state necessarily grows with the trajectory, creating long-horizon dependencies that conventional token-level RL struggles to optimize.

- **Action ($a_t$):** An action is a full textual segment generated by the policy $\pi_\theta$, consisting of free-form reasoning followed by a decision. The action terminates either with a `</tool_call>` token, which triggers a search, putting the results within `</tool_response>`, or with a `</answer>` token, which ends the trajectory. This structured action space forces the agent to learn not only *what* to generate but also *when* to search—an aspect that introduces significant variability in trajectory length.

- **Policy ($\pi_\theta$):** The policy $\pi_\theta$ is the underlying LLM, generating tokens autoregressively conditioned on the state. The policy conditions on the full history, but the optimization target is calculated only on the model-generated thoughts and actions, masking out the retrieved content from the environment (Jin et al., 2025b). Optimizing this policy requires credit assignment over long sequences in which many intermediate reasoning steps do not receive direct supervision or reward, further emphasizing the need for stable sequence-level optimization.

- **Trajectory ($\tau$):** A trajectory $\tau = (s_1, a_1, \ldots, s_T, a_T)$ records all reasoning and tool interactions taken for a given question, including both model-generated actions and environment-returned search results. Because the final reward is assigned at the level of the entire trajectory, the optimization problem is fundamentally sequence-level: every early decision can influence the eventual answer correctness.

- **Reward ($R(\tau)$):** We employ a sparse terminal reward: a trajectory receives $R = 1$ if the final answer contains the ground-truth text, and $R = 0$ otherwise:

$$R(\tau) = \begin{cases} 1 & \text{if } a_{\text{gold}} \subseteq a_{\text{pred}}, \\ 0 & \text{otherwise.} \end{cases} \tag{4}$$

**RL with a Search Engine.** Following Jin et al. (2025b), we explicitly model the search engine, denoted as $\mathcal{S}$, as part of the environment. The policy LLM $\pi_\theta$ learns to generate trajectories by interleaving reasoning with calls to $\mathcal{S}$. The overall optimization problem is to find a policy that maximizes the expected reward, regularized by a KL divergence term to prevent large deviations from a reference policy $\pi_{\text{ref}}$:

$$\max_{\pi_\theta} \mathbb{E}_{x \sim \mathcal{D}, y \sim \pi_\theta(\cdot|x;\mathcal{S})} [R(x,y)] - \beta D_{\text{KL}} [\pi_\theta(y|x;\mathcal{S})||\pi_{\text{ref}}(y|x;\mathcal{S})]. \tag{5}$$

Here, $y \sim \pi_\theta(\cdot|x;\mathcal{S})$ signifies that the trajectory $y$ is generated through a multi-step process involving both the policy's token generation and the information returned by the search engine $\mathcal{S}$.

**Motivation for DSPO.** While frameworks like Search-R1 (Jin et al., 2025b) have successfully framed agentic search as an RL problem, applying conventional algorithms like PPO or GRPO faces significant hurdles. The open-ended nature of the search environment exacerbates the instability of token-level optimization. A core issue is the fundamental mismatch between the unit of sequence-level reward assignment and the unit of token-level optimization (Zheng et al., 2025). This discrepancy leads to high-variance gradient estimates that accumulate over long trajectories, often culminating in policy collapse. Furthermore, the sparse binary reward signal means many training batches may contain only successful or only unsuccessful trajectories, yielding abnormal advantage and thus providing no learning signal, which drastically reduces sample efficiency (Yu et al., 2025; Liu et al., 2025). DSPO is designed to directly counteract these two critical failure modes.

## 3.2 DYNAMIC-FILTER SEQUENCE-LEVEL POLICY OPTIMIZATION

DSPO introduces two key innovations over prior methods: (1) it performs policy optimization at the sequence level, aligning the training objective with the trajectory-based reward structure, and (2) it incorporates a dynamic filtering mechanism to ensure every training batch provides a high-quality, non-zero learning signal. The entire training process, which integrates these components, is depicted in Figure 1.

### 3.2.1 SEQUENCE-LEVEL POLICY OPTIMIZATION FOR ENHANCED STABILITY

Inspired by GSPO (Zheng et al., 2025), we replace the unstable token-level importance ratio with a theoretically grounded sequence-level counterpart. The sequence-level importance ratio $s_i(\theta)$ for a response $y_i$ is defined as the geometric mean of its token-level ratios:

$$s_i(\theta) = \left( \frac{\pi_\theta(y_i|x)}{\pi_{\theta_{\text{old}}}(y_i|x)} \right)^{\frac{1}{|y_i|}} = \exp\left( \frac{1}{|y_i|} \sum_{t=1}^{|y_i|} \log \frac{\pi_\theta(y_{i,t}|x, y_{i,<t})}{\pi_{\theta_{\text{old}}}(y_{i,t}|x, y_{i,<t})} \right). \tag{6}$$

This length normalization is crucial for reducing variance and ensuring that $s_i(\theta)$ remains within a consistent numerical range regardless of sequence length, which is vital for stable clipping.

**Gradient Analysis.** The gradient analysis below shows why DSPO enhances the stability. The gradient of the token-level GRPO objective (unclipped) scales each token's log-probability gradient by a noisy, token-specific weight $r_{i,t}(\theta)$. In contrast, the gradient of our sequence-level objective scales the average log-probability gradient of the entire sequence by a single, more stable sequence-level weight $s_i(\theta)$:

$$\nabla_\theta J_{\text{GRPO}} \propto \mathbb{E}\left[ \hat{A}_i \cdot \sum_{t=1}^{|y_i|} r_{i,t}(\theta) \nabla_\theta \log \pi_\theta(y_{i,t}| \dots) \right] \tag{7}$$

$$\nabla_\theta J_{\text{DSPO}} \propto \mathbb{E}\left[ \hat{A}_i \cdot s_i(\theta) \cdot \sum_{t=1}^{|y_i|} \nabla_\theta \log \pi_\theta(y_{i,t}| \dots) \right] \tag{8}$$

By applying a single, holistic correction factor to the entire trajectory, DSPO avoids the accumulation of token-level noise that plagues prior methods, leading to fundamentally more stable training. In parallel, the dynamic filtering mechanism guarantees a normal advantage signal $\hat{A}_i$ by constructing training batches from rollout groups that contain both successes and failures, thus preventing wasted samples in sparse-reward environments.

### 3.2.2 DYNAMIC OUTCOME-BASED FILTERING FOR EFFICIENT LEARNING

The sparse binary nature of our reward function poses a challenge for group-based advantage estimation. If all $G$ responses in a group are correct ($R = 1$) or all are incorrect ($R = 0$), the normalized advantage $\hat{A}_i$ becomes zero or undefined. Such batches do not provide a useful gradient signal, wasting computational resources.

---

**Algorithm 1** Dynamic-filter Sequence-level Policy Optimization (DSPO)

---

1: **Input:** Initial policy $\pi_{\theta_0}$, fixed reference policy $\pi_{\text{ref}}$, prompt dataset $\mathcal{D}$, group size $G$, batch size $B$, search tool $\mathcal{R}$.
2: Initialize policy $\pi_\theta \leftarrow \pi_{\theta_0}$.
3: **for** each training step **do**
4:      $\pi_{\theta_{\text{old}}} \leftarrow \pi_\theta$.
5:      Initialize training buffer $\mathcal{B} \leftarrow \emptyset$.
6:      **while** $|\mathcal{B}| < B$ **do**
7:          Sample a prompt $x \sim \mathcal{D}$.
8:          Generate a group of $G$ trajectories $\{y_i\}_{i=1}^G$ using $\pi_{\theta_{\text{old}}}$ and the search tool $\mathcal{R}$.
9:          Compute terminal rewards $\{R_i\}_{i=1}^G = \{\text{ContainsAnswer}(y_i, y_{\text{gold}})\}_{i=1}^G$.
10:          **if** $0 < \sum_{i=1}^G R_i < G$ **then**               ▷ Dynamic outcome-based filtering
11:             Add $(x, \{y_i\}_{i=1}^G, \{R_i\}_{i=1}^G)$ to $\mathcal{B}$.
12:          **end if**
13:      **end while**
14:      **for** each $(x, \{y_i\}, \{R_i\})$ in $\mathcal{B}$ **do**
15:          Compute advantages $\{\hat{A}_i\}_{i=1}^G$ via group normalization of $\{R_i\}$.
16:          Compute sequence-level importance ratios $\{s_i(\theta)\}_{i=1}^G$ using Eq. 6.
17:          Compute the DSPO loss for the group using Eq. 11, applying masks to retrieved tokens.
18:      **end for**
19:      Update policy parameters $\theta$ by taking a gradient step on the total loss from $\mathcal{B}$.
20: **end for**

---

To overcome this, DSPO incorporates a dynamic filtering mechanism inspired by DAPO (Yu et al., 2025). During sampling, we only retain groups of trajectories that contain a mix of successful and unsuccessful outcomes. A group $\{y_i\}_{i=1}^G$ is used for training only if its rewards $\{R_i\}_{i=1}^G$ satisfy:

$$0 < \sum_{i=1}^G R_i < G. \tag{9}$$

This ensures that the reward variance within every training group is non-zero, guaranteeing a meaningful advantage signal. This dynamic selection curates a high-quality dataset for each policy update, transforming a sparse reward problem into a dense and efficient learning signal.

### 3.3 THE DSPO OBJECTIVE AND TRAINING ALGORITHM

By integrating these components, we arrive at the final DSPO objective. For each valid group from the filtered sample space $\mathcal{D}_{\text{filtered}}$, we compute the advantage $\hat{A}_i$ using group-relative normalization:

$$\hat{A}_i = \frac{R_i - \text{mean}(R)}{\text{std}(R) + \delta}, \tag{10}$$

where $\delta$ is a small constant for numerical stability. The policy $\pi_\theta$ is updated by maximizing (we omit the KL divergence term to simplify the presentation of the core objective form):

$$J_{\text{DSPO}}(\theta) = \mathbb{E}_{(x,\{y_i\}) \in \mathcal{D}_{\text{filtered}}} \left[ \frac{1}{G} \sum_{i=1}^G \min\left( s_i(\theta)\hat{A}_i, \text{clip}(s_i(\theta), 1 - \epsilon_{\text{low}}, 1 + \epsilon_{\text{high}})\hat{A}_i \right) \right], \tag{11}$$

where $s_i$ is the sequence-level importance ration defined as the geometric mean of the token-level ratios. We use the decoupled clip for better exploration of the policy(Yu et al., 2025). Crucially, during likelihood calculation, we apply loss masking to all tokens retrieved from the search tool following Jin et al. (2025b). This ensures the model learns to utilize external knowledge for reasoning, not simply to reproduce it. The full training process is detailed in Algorithm 1.

## 4 EXPERIMENTS

In this section, we conduct a series of experiments to empirically validate the effectiveness of our proposed Dynamic-filter Sequence-level Policy Optimization (DSPO) algorithm. Our primary

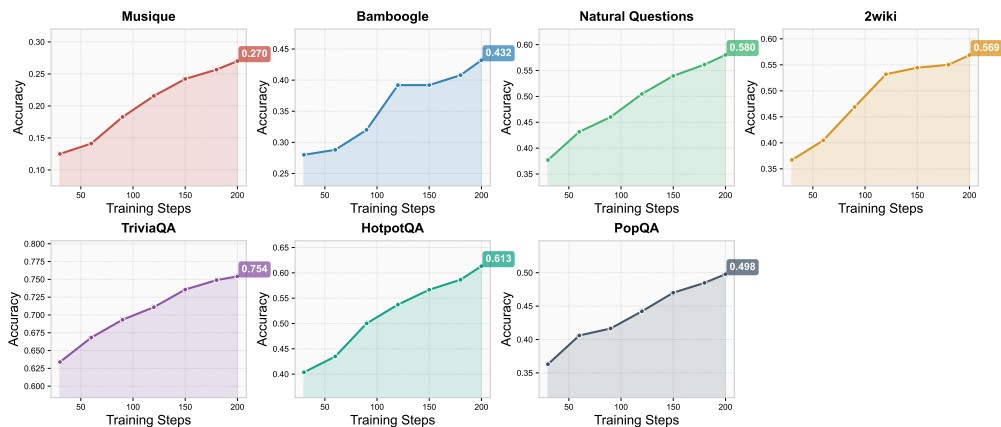

Figure 2: **Validation performance of DSPO across seven benchmarks during training.** The steady, monotonic increase in accuracy confirms that DSPO's reward improvement translates directly to enhanced generalization and that our method learns a robust search-and-reasoning policy.

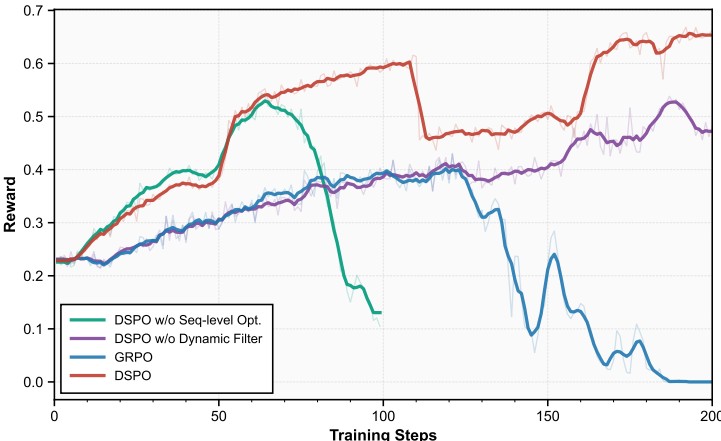

Figure 3: **Training reward dynamics of DSPO and its ablations.** Comparative view of learning curves. DSPO (red) demonstrates stable and monotonic improvement. In contrast, token-level variants (green, blue) suffer catastrophic policy collapse, while the sequence-level variant without our filter (purple) plateaus at a suboptimal level.

objectives are to demonstrate that DSPO: (1) achieves exceptional performance on challenging question-answering benchmarks; (2) exhibits significantly enhanced training stability, avoiding the catastrophic collapse that plagues baseline methods; and (3) derives its performance gains from the synergistic combination of its core components.

## 4.1 EXPERIMENTAL SETUP

**Prompt Template.** Following Search-R1 (Jin et al., 2025b), As shown in Table 1, we use the prompt template to instruct the model's actions during the search task, including `<think>`, `<tool_call>` and `<answer>`.

**Benchmarks and Baselines.** To provide a rigorous evaluation, our experimental design adheres to the established protocol of Search-R1 (Jin et al., 2025b). We train our model on a composite dataset containing the training splits of Natural Questions (NQ) (Kwiatkowski et al., 2019) and HotpotQA (Yang et al., 2018). We then assess its generalization capabilities on the test sets of seven diverse QA benchmarks: NQ, TriviaQA (Joshi et al., 2017), PopQA (Mallen et al., 2022),

---

**Prompt Template.** Answer the given question. You must conduct reasoning inside **`<think>`** and **`</think>`**. first every time you get new information. After reasoning, if you find you lack some knowledge, you can call a search engine by **`<tool_call>`** query **`</tool_call>`** and it will return the top searched results between **`<tool_response>`** and **`</tool_response>`**. You can search as many times as your want. If you find no further external knowledge needed, you can directly provide the answer inside **`<answer>`** and **`</answer>`**, without detailed illustrations. For example, **`<answer>`** Beijing **`</answer>`**. **Question:** ...

---

Table 1: The prompt template used in our experiments.

HotpotQA, 2WikiMultiHopQA (Ho et al., 2020), Musique (Trivedi et al., 2022), and Bamboogle (Press et al., 2022).

Our comparison suite includes strong external baselines and critical internal ablations. External baselines are the Qwen2.5-7B and 14B models trained with PPO and GRPO from the Search-R1 framework (Jin et al., 2025b;a). To deconstruct our method, we also include two internal baselines as ablations: (1) **DSPO w/o dynamic filter**, which is equivalent to GSPO (Zheng et al., 2025), and (2) **DSPO w/o sequence-level opt.**, which reverts to a strong token-level policy, DAPO (Yu et al., 2025).

For implementation, we used `Qwen2.5-7B-Instruct` model as the starting checkpoint for all our training. Our experiments are built upon the VeRL framework (Sheng et al., 2025), for which we adapted the provided `search-r1-like` example code and scripts to suit our methodology. We benchmark DSPO against a comprehensive suite of baselines. For external comparison, we use the PPO and GRPO methods from the Search-R1 framework (Jin et al., 2025b;a). Crucially, as our work utilizes a modified reward function, we retrained these models under our exact experimental conditions to ensure a fair comparison. The results of these retrained models serve as our primary external benchmarks.

**Implementation and Evaluation.** To isolate the benefits of our algorithm, all RL experiments deliberately employ a standard BM25 retriever. This controlled setup ensures that observed performance improvements are directly attributable to the model's learned policy. Across all methods, models are trained using a sparse, binary reward signal based on substring Exact Match (subEM) of the final answer, and subEM serves as the primary evaluation metric.

### 4.2 MAIN RESULTS AND ABLATION STUDY

To provide a holistic view of our algorithm's effectiveness, we present a comprehensive comparison in Table 2. Due to the synergistic nature of DSPO's components, we find it most illustrative to present our main results alongside our ablation study. This single table juxtaposes DSPO against both external state-of-the-art baselines and its own ablated variants, offering a clear and direct assessment of its overall superiority and the indispensability of its core components.

**Comparison with Baselines.** The results in Table 2 underscore DSPO's clear superiority. Our DSPO-trained 7B agent achieves a remarkable average score of 0.531, establishing a new state-of-the-art. This represents a 34.1% relative improvement over the same-sized Search-R1 (GRPO, 7B) model. More strikingly, our 7B agent achieves a slightly better average score than the much larger Search-R1 14B models (both GRPO and PPO). This outcome provides strong evidence that the performance gains stem from a more effective and stable learning algorithm rather than an over-reliance on model scale.

**Analysis of Ablations.** The ablation results, also presented in Table 2, unequivocally demonstrate that both of DSPO's components are indispensable. First, removing the dynamic filter ('w/o dynamic filter', i.e., GSPO) causes a catastrophic drop in performance, with the average score plummeting to 0.313. This highlights its critical role; without the filter, the sequence-level objective is starved of a useful learning signal due to homogeneous-reward batches. Second, ablating sequence-level optimization ('w/o sequence-level opt.', i.e., DAPO) also leads to a significant performance degradation, yielding an average score of 0.406. While this token-level variant outperforms the filter-less

Table 2: Comprehensive comparison of DSPO with baselines and ablation variants on seven QA benchmarks. Baselines include Search-R1 models (7B & 14B) trained with GRPO and PPO (Jin et al., 2025b;a). Ablations remove key components: 'w/o dynamic filter' and 'w/o seq-level opt.'. Original EM scores from Search-R1 are in parentheses. To maintain the consistency of evaluation, we retrained and evaluated them using our adjusted rewards. Best results are in **bold**; second-best are underlined.

| Dataset | Search-R1 | | | | DSPO & Ablations (Ours, 7B) | | |
| | GRPO (7B) | PPO (7B) | GRPO (14B) | PPO (14B) | w/o dyn. filter | w/o seq-lvl opt. | **DSPO** |
| --- | --- | --- | --- | --- | --- | --- | --- |
| NQ | 0.423 (0.429) | (0.393) | 0.535 (0.482) | (0.424) | 0.363 | 0.470 | **0.580** |
| TriviaQA | 0.658 (0.623) | (0.610) | **0.760** (0.667) | (0.660) | 0.515 | 0.695 | 0.754 |
| PopQA | 0.395 (0.427) | (0.397) | 0.477 (0.434) | (0.442) | 0.277 | 0.430 | **0.498** |
| HotpotQA | 0.401 (0.386) | (0.370) | 0.563 (0.429) | (0.436) | 0.330 | 0.438 | **0.613** |
| 2WikiMultiHopQA | 0.357 (0.414) | (0.346) | **0.611** (0.424) | (0.379) | 0.285 | 0.398 | 0.569 |
| Musique | 0.122 (0.162) | (0.146) | 0.260 (0.191) | (0.210) | 0.105 | 0.133 | **0.270** |
| Bamboogle | 0.280 (0.400) | (0.368) | **0.504** (0.492) | (0.480) | 0.288 | 0.280 | 0.432 |
| **Average** | 0.377 (0.396) | (0.385) | 0.530 (0.446) | (0.433) | 0.313 | 0.406 | **0.531** |

one, it falls well short of the full DSPO model. As we show in the next section, it is also prone to catastrophic training instability. This confirms that the synergy is crucial: sequence-level updates are essential for stability, while our dynamic filter is critical for transforming sparse rewards into an efficient learning signal.

Beyond quantitative metrics, we observe that DSPO enables sophisticated search behaviors, including recognize irrelevant results, query reformulation and multi-turn verification (see Appendix A.2 for detailed trajectory examples). All of these behaviors are emerging from pure RL training through DSPO.

### 4.3 ANALYSIS OF TRAINING DYNAMICS

To empirically validate our claims regarding stability and efficiency, we analyze the training reward dynamics of DSPO, its ablations, and key baselines. Figure 3 offers a compelling visualization of these dynamics, reinforcing our core architectural choices.

**DSPO (red)** exhibits a smooth, monotonic ascent, efficiently converging to the highest reward level. This trajectory empirically confirms the stability afforded by its sequence-level objective. In stark contrast, the **token-level methods**—DSPO w/o Seq-level Opt. (green) and vanilla GRPO (blue)—suffer from catastrophic policy collapse early in training. Their rewards plummet after a brief initial improvement, a clear manifestation of the instability caused by high-variance, token-level gradient updates. Meanwhile, **DSPO w/o Dynamic Filter** (purple), which leverages sequence-level updates but lacks an efficient learning signal, remains stable but plateaus at a significantly suboptimal performance ceiling. These dynamics reveal that DSPO's synergy of sequence-level stability and dynamic filtering is key to its robust and effective policy optimization.

To ensure these improvements in training reward translate to genuine generalization rather than reward hacking, we track validation performance on key benchmarks throughout training. As illustrated in Figure 2, DSPO's validation accuracy on NQ, HotpotQA, and other diverse benchmarks rises consistently, mirroring its stable reward curve. This correlation confirms that the agent is learning a generalizable search-and-reasoning policy.

### 4.4 SCALABILITY AND GENERALIZATION ANALYSIS

To further validate the robustness of our approach, we extend our evaluation to explore model scalability and domain generalization.

**Scalability to Larger Models.** We investigate whether the stability benefits of DSPO translate to larger parameter scales by training Qwen2.5-14B-Instruct. As detailed in Table 3, DSPO demonstrates remarkable scalability. The DSPO-trained 14B model achieves an average accuracy of 60.6%,

significantly outperforming the strong GRPO-14B baseline (53.0%) by a relative margin of **14.3%**. These results confirm that our method effectively leverages increased model capacity, establishing an outperforming performance that consistently exceeds standard baselines.

Table 3: Scalability analysis on Qwen2.5-14B-Instruct. Best results are in **bold**.

| Dataset | Instruct (14B) | GRPO (14B) | DSPO (7B) | DSPO (14B) | Gain |
|---------|---------|------|------|-------|-------|
| NQ | 0.345 | 0.535 | 0.580 | **0.629** | +17.6% |
| HotpotQA | 0.407 | 0.563 | 0.613 | **0.665** | +18.1% |
| 2WikiMQA | 0.332 | 0.611 | 0.569 | **0.699** | +14.4% |
| Bamboogle | 0.328 | 0.504 | 0.432 | **0.544** | +7.9% |
| PopQA | 0.364 | 0.477 | 0.498 | **0.545** | +14.3% |
| TriviaQA | 0.643 | 0.760 | 0.754 | **0.802** | +5.5% |
| Musique | 0.151 | 0.260 | 0.270 | **0.361** | +38.8% |
| **Average** | 0.367 | 0.530 | 0.531 | **0.606** | **+14.3%** |

**Generalization to Mathematical Reasoning.** We further assess the universality of DSPO by applying it to single-turn mathematical reasoning tasks using the Qwen2.5 and Qwen3 model family. Table 4 presents the comparison on Math500 and Olympiad-Bench. DSPO consistently surpasses GRPO across both 7B and 4B model sizes. This indicates that DSPO are effective for general reasoning domains.

Table 4: Generalization to mathematical reasoning. Best results are in **bold**.

| Model | Benchmark | Steps | GRPO | DSPO | Gain |
|-------|-----------|-------|------|------|------|
| Qwen2.5-Math-7B | Math500 | 200 | 0.772 | **0.798** | +2.6% |
| Qwen3-4B | Olympiad-Bench | 100 | 0.728 | **0.755** | +2.7% |

## 5 CONCLUSION

In this work, we tackled the critical instability and sample inefficiency issues that plague RL for autonomous LLM search agents. We introduced Dynamic-filter Sequence-level Policy Optimization (DSPO), an improved algorithm that ensures robust training through two key components: sequence-level optimization to prevent catastrophic policy collapse, and a dynamic outcome-based filter to transform sparse rewards into a consistently effective learning signal. Our experiments demonstrated that DSPO not only achieves substantial performance across a suite of challenging question-answering benchmarks but also exhibits superior training stability compared to prior methods.

By enabling robust training from environmental feedback alone, DSPO establishes a practical and efficient blueprint for creating capable LLM agents without costly expert data. With this stable foundation, future work can confidently explore integrating advanced retrievers or extending DSPO to complex, multi-tool tasks. Furthermore, since the challenges of sparse rewards and unstable policy gradients are not unique to search, we hypothesize that DSPO's principles will yield similar performance and stability gains in other domains such as mathematics and code generation, which remains a promising direction for future validation. We believe the core tenets of DSPO—matching the optimization unit to the reward signal and guaranteeing signal density—will be instrumental in developing the next generation of autonomous AI.

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

# A APPENDIX

## A.1 THE USE OF LARGE LANGUAGE MODELS (LLMs)

In adherence to the ICLR 2026 policy, this section details the use of LLMs in the preparation of this manuscript. Their role was significant in enhancing the presentation and accelerating parts of the research process, but not in generating the core scientific ideas. The precise roles are outlined below:

- **Writing and Language Polishing:** A primary use of LLMs was for improving the quality and clarity of the manuscript's text. This included rephrasing sentences for better flow, correcting grammatical errors, suggesting alternative phrasings for technical concepts, and ensuring a consistent academic tone throughout the paper. This iterative process of refinement with the LLM significantly improved the final readability.

- **Literature Retrieval Support:** LLMs assisted in the literature retrieval process by providing summaries of known papers and helping to identify related concepts and terminologies for the background sections. The LLM served as a tool to efficiently explore and summarize the surrounding literature.

- **Code and Visualization Refinement:** For the presentation of our results, LLMs were used to refine the LaTeX code for figures and tables. For instance, the model assisted in iterating on the design and implementation of Table A.1, which presents qualitative trajectory examples, to enhance its visual clarity and professional appearance.

Crucially, the core scientific contributions—including the conceptualization and formulation of the DSPO algorithm, the experimental design, and the analysis of the results—are entirely the original work of the human authors. All content, including text and code generated by the LLM, was meticulously reviewed, critically evaluated, and edited by the authors. We take full responsibility for the entirety of the paper's content, its scientific accuracy, and the originality of its contributions. LLMs were not used in a capacity that would warrant authorship.

## A.2 QUALITATIVE ANALYSIS.

To provide insight into the learned search strategies, Table A.1 presents representative trajectories from our DSPO-trained agent. These examples demonstrate the model's ability to: (1) reformulate queries when initial searches yield irrelevant results, (2) verify information through follow-up searches, and (3) extract comprehensive answers from search results. Notably, the agent exhibits sophisticated reasoning about search quality, as seen in Example 1 where it recognizes off-topic results and autonomously decides to refine its query. This behavior emerges purely from RL training without any supervised demonstrations of search strategies.

Table A.1: Representative trajectories demonstrating DSPO agent's learned search and reasoning capabilities. The agent exhibits sophisticated behaviors including **(a)** recognition of irrelevant results, **(b)** query reformulation, and **(c)** verification strategies, all emerging from pure RL training without supervised demonstrations.

| **Example 1: Presidential Succession** *(Query Refinement after Failure)* | |
|---|---|
| **Question** | Who is next in line if something happens to the president? |
| **Gold** | `vice president` |
| **Turn 1** | `search("who is next in line if something happens to the president")` 
 → *Retrieved: Documents about novels "Something Happened" and "Mosquitoes Buzz"* 
 **Agent:** "The queries were not directly about the line of succession... The relevant information I need is not present in the search results. Let's try searching again with a **more specific query**." |
| **Turn 2** | `search("line of succession for the president")` 
 → *Retrieved: Philippine and US presidential succession information* 
 **Agent:** "Based on the search results... the next person in line would be the Vice President." |
| **Answer** | ✓ The Vice President |

| **Example 2: University Identification** *(Information Verification)* | |
|---|---|
| **Question** | Which American University is situated in Ithaca, New York? |
| **Gold** | `Cornell University` |
| **Turn 1** | `search(["American University in Ithaca", "University in Ithaca NY"])` 
 → *Retrieved: "Ithaca is home to Cornell University, an Ivy League school..."* 
 **Agent:** "Ithaca, New York is home to Cornell University, which is an Ivy League school. However... it is highly likely that the university in question is Cornell. **Let me confirm this**." |
| **Turn 2** | `search(["Cornell University location", "Cornell University New York"])` 
 → *Retrieved: Cornell history confirming establishment in Ithaca* 
 **Agent:** "Cornell University is indeed located in Ithaca, New York." |
| **Answer** | ✓ Cornell University |

| **Example 3: Biographical Information** *(Direct Extraction)* | |
|---|---|
| **Question** | What is John Andrew Martin's occupation? |
| **Gold** | `lawyer, attorney, politician` |
| **Turn 1** | `search(["John Andrew Martin occupation", "John Andrew Martin job"])` 
 → *Retrieved: "American journalist, attorney, soldier, and politician..."* 
 **Agent:** "John Andrew Martin was an American journalist, attorney, soldier, and politician. He represented Colorado in the U.S. House of Representatives." |
| **Answer** | ✓ Journalist, attorney, soldier, and politician |

