# OpenReview forum: "DSPO: Stable and Efficient Policy Optimization for Agentic Search and Reasoning"
_ICLR.cc/2026/Conference — Submitted to ICLR 2026_

### Official Review · Reviewer_oBxZ · 2025-10-23

**Soundness:** 3
**Presentation:** 3
**Contribution:** 3
**Rating:** 4
**Confidence:** 4

**Summary:**

This paper proposes DSPO, an improved RL algorithm that overcomes the core instability and sample-inefficiency issues in training agentic search models. It achieves this by unifying two key principles into a single cohesive framework: sequence-level optimization for robust policy updates and dynamic outcome-based filtering for a dense and effective learning signal.

**Strengths:**

1, Clear synthesis of sequence-level optimization with dynamic outcome filtering into a single, coherent algorithm tailored to agentic search with sparse terminal rewards. The sequence-level clipping with length-normalized ratio is well motivated and derived.
2, Paper is well written and ablation studies do make sense to me.

**Weaknesses:**

See questions

**Questions:**

1, Could you provide the results from different model family such as Llama3 to see how generalize the proposed method is?
2, Could you also provide the scaled results for your proposed method on 14B models? I can get that the performance of 7B model is already good but I think it would be better to show your method scalability by using different size models from the same model family

---

> ### Author Response · Authors · 2025-12-04
> **To Reviewer oBxZ**
>
> We thank you for your insightful review. Below we address the concerns.
>
> **Q1: Could you provide the results from different model family such as Llama3 to see how generalize the proposed method is? 2, Could you also provide the scaled results for your proposed method on 14B models? I can get that the performance of 7B model is already good but I think it would be better to show your method scalability by using different size models from the same model family**
>
> A1: Thank you for your suggestions! We fully agree with the importance of validating DSPO on different model families. While we did not perform experiments specifically on Llama 3 due to the limited computation resource and time window, we'll explore this in the future.
>
> And we successfully scaled DSPO to the Qwen2.5-14B-Instruct model. The 14B model with DSPO significantly outperforms both the The 14B model with GRPO and our own 7B results. This confirms that DSPO scales effectively to larger sizes.
>
> | Dataset | Instruct (14B) | GRPO (14B) | DSPO (7B) | **DSPO (14B)** | **DSPO vs. GRPO 14B** |
> | :--- | :---: | :---: | :---: | :---: | :---: |
> | **NQ** | 0.345 | 0.535 | 0.580 | **0.629** | **+17.6%** |
> | **HotpotQA** | 0.407 | 0.563 | 0.613 | **0.665** | **+18.1%** |
> | **2WikiMQA** | 0.332 | 0.611 | 0.569 | **0.699** | **+14.4%** |
> | **Bamboogle** | 0.328 | 0.504 | 0.432 | **0.544** | **+7.9%** |
> | **PopQA** | 0.364 | 0.477 | 0.498 | **0.545** | **+14.3%** |
> | **TriviaQA** | 0.643 | 0.760 | 0.754 | **0.802** | **+5.5%** |
> | **MuSiQue** | 0.151 | 0.260 | 0.270 | **0.361** | **+38.8%** |
> | **Average** | **0.367** | **0.530** | **0.531** | **0.606** | **+14.3%** |

---

### Official Review · Reviewer_Wt3z · 2025-10-31

**Soundness:** 2
**Presentation:** 2
**Contribution:** 2
**Rating:** 4
**Confidence:** 4

**Summary:**

This paper introduces Dynamic-filter Sequence-level Policy Optimization (DSPO), a novel RL algorithm designed to train LLMs to act as stable and effective autonomous agents for search and reasoning tasks. The authors identify two critical failures in existing RL methods: training instability caused by a mismatch between token-level optimization and sequence-level rewards, and inefficient learning from sparse rewards where training batches often lack a useful learning signal. DSPO addresses these issues with two key innovations: first, it employs sequence-level policy optimization to align the training objective with the overall trajectory reward, which fundamentally stabilizes the learning process. Second, it incorporates a dynamic outcome-based filtering mechanism that ensures every training batch contains a mix of both successful and unsuccessful outcomes, guaranteeing a consistent and effective advantage signal. Through experiments, the paper demonstrates that DSPO not only prevents the catastrophic policy collapse common in other methods but also achieves state-of-the-art performance, with a 7B model outperforming even a 14B baseline model on complex question-answering benchmarks.

**Strengths:**

1. Optimize the problems of GRPO in multi-round agent interaction scenarios
2. Clear logic

**Weaknesses:**

1、The most significant issue with this paper lies in its experimental evaluation. The comparison is limited to the GRPO algorithm, overlooking numerous recent improvements such as DAPO and GSPO. These methods also tackle challenges related to token-level advantage calculation and gradient updates, yet the authors have not benchmarked their proposed algorithm against them.

2、In the experiments, the authors only test the effectiveness of DSPO on search tasks and do not evaluate it on a broader range of tasks, such as mathematics. I am curious how DSPO would perform compared to GRPO in single-turn interaction scenarios, like those found in mathematical reasoning tasks.

3、The final point concerns novelty. The second contribution proposed by the authors—filtering samples based on the reward distribution of group rollouts, such as selecting for groups with high reward variance—is very similar to existing methods. The authors need to clarify the specific innovation of their approach compared to prior work.

**Questions:**

Stated in Weaknesses

---

> ### Author Response · Authors · 2025-12-04
> **To Reviewer Wt3z**
>
> We thank you for your insightful review. Below we address the concerns.
>
> **Q1: The most significant issue with this paper lies in its experimental evaluation. The comparison is limited to the GRPO algorithm, overlooking numerous recent improvements such as DAPO and GSPO. These methods also tackle challenges related to token-level advantage calculation and gradient updates, yet the authors have not benchmarked their proposed algorithm against them.**
>
> A1: Thanks! We chose PPO and GRPO because they are the foundational and most widely used algorithms for LLM RL. Since DSPO is a direct improvement over the optimization objective of these methods, they are the most rigorous comparisons. Our ablation studies effectively serve as comparisons against the individual methods. The "w/o filter" setting represents a standard Sequence-level approach (like GSPO), and "w/o seq-level" represents a Filtering-only approach (like a big part of DAPO). And we use the decoupled clip from DAPO for all algorithm settings in our experiments.
>
> **Q2: In the experiments, the authors only test the effectiveness of DSPO on search tasks and do not evaluate it on a broader range of tasks, such as mathematics. I am curious how DSPO would perform compared to GRPO in single-turn interaction scenarios, like those found in mathematical reasoning tasks.**
>
> A2: Thank you for the suggestion. To demonstrate the generalizability of DSPO beyond agentic search, we conducted additional experiments on mathematical reasoning tasks using the Qwen2.5-Math-7B and Qwen-3-4B with math_verify as the verifier. As shown in the table below, DSPO consistently outperforms GRPO on both Math500 and Olympiad-Bench. We'll add these detailed results to Appendix.
>
> | Model | Benchmark | Steps | GRPO (Acc) | **DSPO (Acc)** | **Improvement** |
> | :--- | :--- | :--- | :--- | :--- | :--- |
> | **Qwen2.5-Math-7B** | Math500 | 200 | 77.2% | **79.8%** | **+2.6%** |
> | **Qwen2.5-4B** | Olympiad-Bench | 100 | 72.8% | **75.5%** | **+2.7%** |
>
> **Q3: The final point concerns novelty. The second contribution proposed by the authors—filtering samples based on the reward distribution of group rollouts, such as selecting for groups with high reward variance—is very similar to existing methods. The authors need to clarify the specific innovation of their approach compared to prior work.**
>
> A3: Thanks! We acknowledge that GSPO and DAPO are established concepts. However, our core contribution lies in identifying that neither component alone is sufficient for the challenging task of agentic search. Our contribution is the empirical discovery and the recipe for stable and high performance agentic search RL.

---

### Official Review · Reviewer_fdXC · 2025-11-01

**Soundness:** 2
**Presentation:** 2
**Contribution:** 2
**Rating:** 2
**Confidence:** 3

**Summary:**

This paper introduces DSPO (Dynamic-filter Sequence-level Policy Optimization), a new Reinforcement Learning (RL) algorithm designed to stably and efficiently train Large Language Models (LLMs) for tasks requiring agentic search and reasoning. The authors identify two main problems with existing methods like GRPO: Instability, where a mismatch between sequence-level rewards and token-level optimization objectives leads to high-variance gradients and policy collapse; and Inefficiency, where sparse rewards often lead to training batches where all trajectories fail or all succeed, providing no useful learning signal. DSPO tackles this by combining two key techniques into a single framework: Sequence-Level Optimization, which aligns the optimization objective with the reward unit by optimizing at the full-sequence level for a more stable gradient, and Dynamic Outcome-Based Filtering, which actively filters training batches to ensure every batch contains a mix of both successful and unsuccessful trajectories, guaranteeing an effective learning signal. The paper shows that their DSPO-trained 7B model significantly outperforms a 7B baseline (by 34.1%) and even surpasses a 14B baseline model on the complex HotpotQA benchmark, all while demonstrating superior training stability.

**Strengths:**

1. The paper does an excellent job of explaining why training agentic search models with RL is difficult, pinpointing the specific issues of objective mismatch and sparse reward inefficiency.

2. The performance looks good. A 7B model outperforming a 14B model on a complex multi-hop QA task (HotpotQA) is a very strong and compelling result.

3. The appendix (Table A.1) provides concrete examples of the agent's learned behavior, such as recognizing irrelevant search results and reformulating queries, which shows it's learning a genuinely useful policy.

**Weaknesses:**

1. The primary external baselines (GRPO and PPO) are from a single framework (Search-R1). While the authors do the right thing by re-training these models under their conditions for a fair comparison, they don't compare against a wider variety of recent RL-for-LLM algorithms. The paper builds on ideas from GSPO and DAPO, but the ablations ("w/o filter" and "w/o seq-level") act as stand-ins rather than a direct comparison against the original, fully-tuned implementations of those methods.

2. The DSPO algorithm itself seems general, but its empirical validation is limited to the agentic search setting from the Search-R1 paper. It's unclear how it would perform on other complex RL tasks for LLMs, such as math reasoning or code generation, which also suffer from similar stability and reward issues.

3.  All the evaluation datasets—2WikiMQA, HotpotQA, Bamboogle, MuSiQue, NQ, TriviaQA, and PopQA—are quite old, and the models have likely already been exposed to them during pretraining. Prior studies have also shown that models can often find the correct answer with only a few search turns (≤1). It would strengthen the evaluation to include more challenging and up-to-date datasets.

4. The core contribution is a synthesis of two existing ideas (sequence-level optimization from GSPO and dynamic filtering from DAPO). While the paper proves their combination is synergistic and necessary, it does lower the fundamental novelty of the algorithm itself. After GSPO solved the core stability problem, adding a filtering mechanism could be seen as an incremental, though effective, step.

**Questions:**

1. What is the computational and sample overhead of the dynamic filtering? What percentage of generated trajectory groups are discarded, and how does this change over the course of training?

2. How does DSPO's performance (and the learned policy) change when it is paired with a state-of-the-art dense retriever instead of BM25?

3. The paper shows a 7B model beating a 14B baseline. What are the results when DSPO is applied to the 14B model itself? Do the stability and performance gains scale?

---

> ### Author Response · Authors · 2025-12-04
> **To Reviewer fdXC (1/2)**
>
> We thank you for your insightful review. Below we address the concerns.
>
> **W1: The primary external baselines (GRPO and PPO) are from a single framework (Search-R1). While the authors do the right thing by re-training these models under their conditions for a fair comparison, they don't compare against a wider variety of recent RL-for-LLM algorithms. The paper builds on ideas from GSPO and DAPO, but the ablations ("w/o filter" and "w/o seq-level") act as stand-ins rather than a direct comparison against the original, fully-tuned implementations of those methods.**
>
> A1: Thanks! We chose PPO and GRPO because they are the foundational and most widely used algorithms for LLM RL. Since DSPO is a direct improvement over the optimization objective of these methods, they are the most rigorous comparisons. Our ablation studies effectively serve as comparisons against the individual methods. The "w/o filter" setting represents a standard Sequence-level approach (like GSPO), and "w/o seq-level" represents a Filtering-only approach (like a big part of DAPO). And we use the decoupled clip from DAPO for all algorithm settings in our experiments.
>
> **W2: The DSPO algorithm itself seems general, but its empirical validation is limited to the agentic search setting from the Search-R1 paper. It's unclear how it would perform on other complex RL tasks for LLMs, such as math reasoning or code generation, which also suffer from similar stability and reward issues.**
>
> A2: We appreciate this suggestion. To demonstrate the generalizability of DSPO beyond agentic search, we conducted additional experiments on mathematical reasoning tasks using the Qwen2.5-Math-7B and Qwen-3-4B with math_verify as the verifier. As shown in the table below, DSPO consistently outperforms GRPO on both Math500 and Olympiad-Bench. We'll add these detailed results to Appendix.
>
> | Model | Benchmark | Steps | GRPO (Acc) | **DSPO (Acc)** | **Improvement** |
> | :--- | :--- | :--- | :--- | :--- | :--- |
> | **Qwen2.5-Math-7B** | Math500 | 200 | 77.2% | **79.8%** | **+2.6%** |
> | **Qwen2.5-4B** | Olympiad-Bench | 100 | 72.8% | **75.5%** | **+2.7%** |
>
> **W3: All the evaluation datasets—2WikiMQA, HotpotQA, Bamboogle, MuSiQue, NQ, TriviaQA, and PopQA—are quite old, and the models have likely already been exposed to them during pretraining. Prior studies have also shown that models can often find the correct answer with only a few search turns (≤1). It would strengthen the evaluation to include more challenging and up-to-date datasets.**
>
> A3: Thanks. We acknowledge the validity of the concern regarding dataset age and the potential for pre-training exposure. And we respectfully argue that memorization may not explain the results. Prior works like Search-R1 demonstrate that direct inference or standard CoT on these datasets yields lower accuracy compared to the trained models. The performance gap suggests that the model is relying on the learned search and reasoning policy rather than simple knowledge recall. And we'll try to use more challenging and up-to-date datasets to further validate our work.
>
> **W4: The core contribution is a synthesis of two existing ideas (sequence-level optimization from GSPO and dynamic filtering from DAPO). While the paper proves their combination is synergistic and necessary, it does lower the fundamental novelty of the algorithm itself. After GSPO solved the core stability problem, adding a filtering mechanism could be seen as an incremental, though effective, step.**
>
> A4: Thanks! We acknowledge that GSPO and DAPO are established concepts. However, our core contribution lies in identifying that neither component alone is sufficient for the challenging task of agentic search. Our contribution is the empirical discovery and the recipe for stable and high performance agentic search RL.
>
> **Q1: What is the computational and sample overhead of the dynamic filtering? What percentage of generated trajectory groups are discarded, and how does this change over the course of training?**
>
> A1: Thanks. We acknowledge that DSPO involves a degree of oversampling (generating more trajectories than are used in the update) to ensure mixed batches. It's a efficient trade-off to make every step useful. And the discard rate is dynamic according to the model's capacity. We'll include detailed statistics on discard rates for future work.
>
> **Q2: How does DSPO's performance (and the learned policy) change when it is paired with a state-of-the-art dense retriever instead of BM25?**
>
> A2: Thanks. Using a dense retriever would likely raise the upper bound of performance for all methods. But we use the simple one for our training. This rigorously tests the robustness of the learned policy—its ability to filter out irrelevant information and reformulate queries when the retriever fails. Showing strong performance with a weak retriever highlights the robustness of the DSPO policy. And we'll test the dense one for better representation.

---

> ### Author Response · Authors · 2025-12-04
> **To Reviewer fdXC (2/2)**
>
> **Q3: The paper shows a 7B model beating a 14B baseline. What are the results when DSPO is applied to the 14B model itself? Do the stability and performance gains scale?**
>
> A3: Thanks! We use Qwen2.5-14B-Instruct for the init model and train with the same settings using DSPO (step 200), the results are as follows:
>
> | Dataset | Instruct (14B) | GRPO (14B) | DSPO (7B) | **DSPO (14B)** | **DSPO vs. GRPO 14B** |
> | :--- | :---: | :---: | :---: | :---: | :---: |
> | **NQ** | 0.345 | 0.535 | 0.580 | **0.629** | **+17.6%** |
> | **HotpotQA** | 0.407 | 0.563 | 0.613 | **0.665** | **+18.1%** |
> | **2WikiMQA** | 0.332 | 0.611 | 0.569 | **0.699** | **+14.4%** |
> | **Bamboogle** | 0.328 | 0.504 | 0.432 | **0.544** | **+7.9%** |
> | **PopQA** | 0.364 | 0.477 | 0.498 | **0.545** | **+14.3%** |
> | **TriviaQA** | 0.643 | 0.760 | 0.754 | **0.802** | **+5.5%** |
> | **MuSiQue** | 0.151 | 0.260 | 0.270 | **0.361** | **+38.8%** |
> | **Average** | **0.367** | **0.530** | **0.531** | **0.606** | **+14.3%** |
>
> As shown in the table above, scaling DSPO to 14B yields substantial performance gains. The 14B model with DSPO not only significantly outperforms the the one with GRPO across all datasets but also establishes a better result compared to our previous 7B model.

---

### Official Review · Reviewer_skPz · 2025-11-08

**Soundness:** 3
**Presentation:** 3
**Contribution:** 3
**Rating:** 6
**Confidence:** 4

**Summary:**

The paper proposes DSPO (Dynamic-filter Sequence-level Policy Optimization), an RL algorithm for training agentic LLMs to interleave multi-turn search and reasoning without supervised demonstrations. DSPO combines (i) sequence-level optimization (GSPO) to align optimization with sequence-level rewards, and (ii) dynamic outcome-based filtering (DAPO) to ensure batches contain mixed success/failure trajectories, stabilizing group-relative advantages. On seven QA benchmarks with a BM25 retriever, a DSPO-trained 7B model reportedly outperforms a comparable 7B baseline by 34.1% (relative) and even exceeds a 14B baseline on HotpotQA. The paper emphasizes stability, sample efficiency, and generalization.

**Strengths:**

1.	This paper solves the problem of mismatch between sequence-level rewards and token-level optimization (GRPO/PPO), and sparse rewards leading to homogeneous groups and collapsed advantages.
2.	The principle of the proposed algorithm is very clear: combining sequence-level ratios (geometric mean, length-normalized) with dynamic filtering directly targets instability and sparsity.
3.	The results demonstrate notable advantage of DSPO’s performance on 7 datasets, with only BM25. 7B DSPO beats 14B baselines on HotpotQA.
4.	The ablation study supports their claim: Removing dynamic filtering (GSPO) or sequence-level optimization (DAPO) degrades performance; the pattern aligns with the hypothesized failure modes.

**Weaknesses:**

1.	Some mathematical definitions and formulations are not very clear. What is the exact reference policy and KL term in the final loss. The KL regularization omits it in the final objective. The reward and environment are denoted as R. The definition of sequence and trajectory.
2.	Advantage computation: Eq. 10 uses group-wise normalization with std(R)+δ for binary rewards; with small G, variance estimates are noisy. Sensitivity to G and δ is not discussed. Also, what is the exact reference policy and KL term in the final loss (Eq. 11 does not include KL, unlike Eq. 2)? The text mentions KL regularization earlier but omits it in the final objective.
3.	Some details are not mentioned: retrieved tokens is not detailed, the choice of  reference models, initialization checkpoint specifics and instruction format.
4.	Hyperparameters (β for KL if used, δ, group size G, batch size B, learning rate, optimizer, entropy bonus), rollout depth limits, and early stopping criteria are not specified.
5.	Reference model choice and freezing, as well as initialization checkpoint specifics and instruction format, are lightly described.
6.	The filter rejects all-0 or all-1 groups, which changes the data distribution for policy updates. While it guarantees variance for A-^, it can bias the optimization toward decision boundaries, potentially harming calibration or robustness.
7.	There is no analysis showing how filtering impacts exploration (e.g., encouraging diverse query reformulations vs. premature exploitation).

**Questions:**

1.	The content of “Agentic Search as a Markov Decision Process” is mainly a introduction of MDP, and should be rewritten.
2.	In fig.3, what do thick curves and thin curves stand for? Why are the thin one so steep？why the green one (DSPO w/o Seq-level Opt.) only has 100 steps? Why the red one (DSPO) has a decease from step 110 to 160?
3.	The baselines are only PPO, GRPO? Why DAPO is not compared? More related methods should be considered. Although seven datasets are considered, experiments seem a little insufficient.
4.	The impact of δ should be discussed.

---

> ### Author Response · Authors · 2025-12-04
> **To Reviewer skPz (1/2)**
>
> Thank you for your insightful feedback! Below we address the specific concerns.
>
> **Q1: Some mathematical definitions and formulations are not very clear. The KL term, definition of reward and environment, and definition of sequence and trajectory.**
>
> A1: Thank you for your suggestions! We acknowledge that in Eq. 11, we omitted the KL divergence term to simplify the presentation of the core gradient form. However, the KL constraint is included in our implementation and experiments. We'll add the explanation in our paper.
>
>  We have revised the description to distinguish the concepts. We now denote the environment (Search Engine here) simply as $\mathcal{S}$, and $\mathcal{R}$ specifically refers to the outcome reward (0 or 1) provided by the environment upon task completion.
>
> We clarified that a trajectory $\tau$ includes the full history (including search results), while our optimization targets the generated sequence. And we'll state that we apply a loss mask to environment-returned search results.
>
> **Q2: Advantage computation: Eq. 10 uses group-wise normalization with std(R)+δ for binary rewards; with small G, variance estimates are noisy.**
>
> A2: In our experiments we set the group size $G=5$, which represents a trade-off between computational efficiency and performance, consistent with settings in prior works. We acknowledge that we did not perform a comprehensive sensitivity analysis for G and δ due to computational constraints. The parameter δ is solely for numerical stability to prevent division by zero. In our experiments, the method is robust to δ as long as it is sufficiently small. We adopted these common standard values to establish the solid settings. Our significant performance gains under these settings demonstrate the robustness of our proposed algorithm.
>
> **Q3: Some details are not mentioned: retrieved tokens is not detailed, the choice of reference models, initialization checkpoint specifics and instruction format.**
>
> A3: Thank you for pointing out the omission. We'll add a section in the Appendix. To clarify here:
>
> - Retrieved Tokens: We use the top-3 documents returned by BM25, truncated to fit the context window, and put them into `<tool_response>` tag.
> - Reference Model: We use a frozen copy of the Instruct model as the reference policy.
> - Initialization: The policy is initialized from the Instruct model (Qwen2.5-7B-Instruct here).
> - Instruction Format: We'll add the prompt template in the paper.
>
> **Q4: Hyperparameters (β for KL if used, δ, group size G, batch size B, learning rate, optimizer, entropy bonus), rollout depth limits, and early stopping criteria are not specified.**
>
> A4: Thank you for pointing out! We'll soon include a detailed hyperparameter table in the Appendix of the revised paper.
>
> **Q5: Reference model choice and freezing, as well as initialization checkpoint specifics and instruction format, are lightly described.**
>
> A5: Thanks! Please refer to our response to Weakness 3. The reference model is identical to the initial policy (Instruct model) and remains frozen during training to compute the KL penalty.
>
> **Q6: The filter rejects all-0 or all-1 groups, which changes the data distribution for policy updates. While it guarantees variance for A-^, it can bias the optimization toward decision boundaries, potentially harming calibration or robustness.**
>
> A6: Thanks! TWe appreciate this insightful concern. We respectfully argue that this filtering acts as a denoising mechanism rather than a harmful bias. The filtering mechanism automatically rejects samples that are currently too hard or too easy. This forces the optimization to focus on the decision boundary where the model is capable of both success and failure. This creates a curriculum that adapts to the model's evolving capabilities, ensuring efficient learning from sparse rewards.
>
> **Q7: There is no analysis showing how filtering impacts exploration (e.g., encouraging diverse query reformulations vs. premature exploitation).**
>
> A7: Thanks! We clarify that filtering is a post-sampling operation, not a pre-sampling constraint. Exploration happens during the rollout phase (controlled by temperature and sampling). The model does explore on all queries. Filtering simply decides which of those explored trajectories provide a valid learning signal. Therefore, filtering does not hinder exploration.

---

> ### Author Response · Authors · 2025-12-04
> **To Reviewer skPz (2/2)**
>
> **Q1: The content of “Agentic Search as a Markov Decision Process” is mainly a introduction of MDP, and should be rewritten.**
>
> A1: Thank you for pointing out! We have rewritten the content for clarity.
>
> **Q2: In fig.3, what do thick curves and thin curves stand for? Why are the thin one so steep？why the green one (DSPO w/o Seq-level Opt.) only has 100 steps? Why the red one (DSPO) has a decease from step 110 to 160?**
>
> A2: Thanks! Thin curves represent raw training metrics (which are naturally noisy in RL), while thick curves are smoothed using Exponential Moving Average (EMA) for clarity.
>
> The green curve represents the ablation "w/o Sequence-level Optimization". This run stopped early because the model suffered from policy collapse, causing the training to become unstable and terminate. This highlights the necessity of our proposed Sequence-level Optimization.
>
> The dip observed from step 110 corresponds to the start of the second training epoch. This is a normal transient fluctuation caused by the epoch transition, where the model re-encounters training prompts and samples new trajectories with the updated policy. As shown in the figure, the curve quickly recovers and continues to improve, confirming that this is a temporary training dynamic rather than instability or collapse (unlike the green curve).
>
> **Q3: The baselines are only PPO, GRPO? Why DAPO is not compared? More related methods should be considered. Although seven datasets are considered, experiments seem a little insufficient.**
>
> A3: Thanks! DAPO is actually a component of our proposed method (the filtering part). The ablation study effectively represents the performance of using only the filtering mechanism (DAPO) without the sequence-level optimization.
>
> We chose PPO and GRPO because they are the foundational and most widely used algorithms for LLM RL. Since DSPO is a direct improvement over the optimization objective of these methods, they are the most rigorous comparisons. And we'll try to use more datasets to further validate our work.
>
> **Q4: The impact of δ should be discussed.**
>
> A4: Thanks! As mentioned in Weakness 2, δ is a small constant used for numerical stability. We'll discuss the impact in the future.

---

### Meta-Review · Area_Chair_mnoZ · 2026-01-07

**Summary:**

While the author rebuttal has addressed many of the reviewers’ concerns, several issues remain that could substantially affect the final rating of the submission:

- The evaluation relies on relatively old benchmarks. While improvements on these datasets are still meaningful, it would be important to additionally evaluate on more recent data that (i) involve a larger number of search turns and (ii) do not provide in-domain training data. Such evaluations would better demonstrate the value of the proposed approach to the community.

- The experiments are conducted exclusively on the Qwen model family, which limits the generality of the conclusions.

- Further discussion is needed to more clearly address the reviewers’ concerns regarding the novelty of the proposed method.

**Reviewer Concerns:**

The last two reviewers will not change their mind.

**Reviewer Scores:**

The last two reviewers will not change their mind. The others may rise the scores but that cannot solve the remaining issues listed above.

---

### Decision · Program_Chairs · 2026-01-26

Reject